# Reduced auditory cortical adaptation in autism spectrum disorder

Rachel Millin[1], Tamar Kolodny[1], Anastasia V Flevaris[1], Alexander M Kale[1], Michael-Paul Schallmo[1], Jennifer Gerdts[2], Raphael A Bernier[2], Scott Murray[1]*

[1]Department of Psychology, University of Washington, Seattle, United States; [2]Department of Psychiatry and Behavioral Sciences, University of Washington, Seattle, United States

**Abstract** Adaptation is a fundamental property of cortical neurons and has been suggested to be altered in individuals with autism spectrum disorder (ASD). We used fMRI to measure adaptation induced by repeated audio-visual stimulation in early sensory cortical areas in individuals with ASD and neurotypical (NT) controls. The initial transient responses were equivalent between groups in both visual and auditory cortices and when stimulation occurred with fixed-interval and randomized-interval timing. However, in auditory but not visual cortex, the post-transient sustained response was greater in individuals with ASD than NT controls in the fixed-interval timing condition, reflecting reduced adaptation. Further, individual differences in the sustained response in auditory cortex correlated with ASD symptom severity. These findings are consistent with hypotheses that ASD is associated with increased neural responsiveness but that responsiveness differences only manifest after repeated stimulation, are specific to the temporal pattern of stimulation, and are confined to specific cortical regions.
DOI: https://doi.org/10.7554/eLife.36493.001

*For correspondence:
somurray@uw.edu

Competing interests: The authors declare that no competing interests exist.

## Introduction

Autism spectrum disorder (ASD) is a behaviorally defined, heterogeneous disorder with significant genotypic and phenotypic complexity. This complexity has made it challenging to identify the underlying neural mechanism(s) that are disrupted in the disorder. However, a unifying theme of numerous proposals is that the pathophysiology of autism is related to a pervasive increase in neural responsiveness (*Rubenstein and Merzenich, 2003*; *Markram and Markram, 2010*; *Gogolla et al., 2009*; *Blatt and Fatemi, 2011*; *Gatto and Broadie, 2010*; *Pizzarelli and Cherubini, 2011*; *Yizhar et al., 2011*; *Hussman, 2001*). However, outside of cases in specific animal models of autism (*Bateup et al., 2011*; *Contractor et al., 2015*; *Bateup et al., 2013*), experimental evidence for increased neural responses has been limited. Studies in humans across a variety of brain regions and measurement techniques have yielded equivocal results, with recent studies finding equivalent (*Dinstein et al., 2012*), or even decreased (*Haigh et al., 2015*) neural responses in subjects with ASDs.

We hypothesized that changes in neural responsiveness in ASD might manifest not in the initial transient neural response but, instead, in how neural responses adapt to repeated stimulation. This hypothesis was motivated by behavioral observations of reduced adaptation in ASD across diverse domains, including tactile stimulation (*Tommerdahl et al., 2007*; *Tannan et al., 2008*; *Puts et al., 2014*), face discrimination (*Pellicano et al., 2007*; *Fiorentini et al., 2012*), gaze direction (*Pellicano et al., 2013*), numerosity (*Turi et al., 2015*), audio-visual asynchrony (*Noel et al., 2017*), and saccade amplitude (*Mosconi et al., 2013*). Further, theoretical work has suggested that many of the core symptoms of ASD, including the failure to adapt or habituate, can be framed as an inability to form predictions (*Sinha et al., 2014*) and incorporate prior experiences (*Pellicano and Burr,*

*2012*). Thus, we tested whether neural adaptation differences in ASD are sensitive to the regularity of the temporal pattern of stimulation. Finally, the specificity of sensory symptoms that occur in individuals with ASD (*Leekam et al., 2007*; *Brown and Dunn, 2002*) suggests that adaptation differences may be modality specific and confined to particular regions of cortex. To test these hypotheses, we used fMRI to characterize neural response amplitudes across time in visual and auditory cortices in adult individuals with high-functioning ASD and neurotypical (NT) control subjects.

## Results

We measured the fMRI response to repeated audio-visual stimulation in early visual and auditory cortical areas in response to brief (200 ms), simultaneous audio (white noise) and visual (checkerboard) stimulus presentations (*Figure 1A*). To help maintain attentional engagement, subjects pressed a button in response to each stimulus presentation. The stimuli were presented in blocks (20 s) with either fixed-interval timing (2 s) or randomized-interval timing (pseudorandom intervals; mean = 2 s; *Figure 1A*). Block order was randomized. Reaction times (RTs) in response to the first stimulus presentation of a block were equivalent for ASD and NT participants (group x timing condition ANOVA; $F_{1,46} = 0.26$, p=0.61; *Figure 1B*). RTs for all other trials were shorter for the ASD participants; using the mean RT over trials 2 – 10 there was a significant main effect of group ($F_{1,46} = 4.84$, p=0.03), a main effect of timing condition ($F_{1,1} = 58.9$, p<0.001), and no interaction ($F_{1,46} = 0.002$; p=0.97). There was no group difference in response errors defined by more than one button press per trial ($F_{1,46} = 0.71$, p=0.13).

For the fMRI data, regions of interest (ROIs) were defined in early visual and auditory cortices based on a statistical contrast between sensory stimulation and rest conditions (*Figure 1C*; see Materials and methods). We defined two distinct temporal epochs in the fMRI response timecourse: the initial transient (4 – 6 s post-onset for the auditory ROI and 6 – 8 s post-onset for the visual ROI) and the sustained response (12 – 20 s post-onset for both auditory and visual ROIs; see gray regions in *Figure 2A and B*). The average timecourses clearly show that the transient response is equivalent for the ASD and NT individuals in both visual and auditory cortices and in both the fixed-interval and randomized-interval timing conditions (*Figure 2A and B*). However, the sustained response – the period of time over which we expected to observe adaptation effects – was markedly elevated in the ASD compared to the NT individuals in auditory cortex (*Figure 2A*). To quantify this effect, we averaged over the timepoints in the sustained period (*Figure 2C*). In the auditory ROI, there was a significant main effect of group (ASD and NT; $F_{1,38} = 8.69$, p=0.005), a trend of timing conditions (fixed- and randomized-interval; $F_{1,1} = 3.51$, p<0.07), and no interaction between the two ($F_{1,38} = 0.295$, p=0.59). Planned comparisons showed that there was only a significant difference between NT and ASD in the fixed-interval condition ($t_{38} = 2.46$, p=0.02, Cohen's d = 0.79) and not in the randomized-interval condition ($t_{38} = 1.58$, p=0.12, Cohen's d = 0.50). In the visual ROI, there were no significant main effects (group, $F_{1,42} = 1.07$, p=0.31; timing condition, $F_{1,1} = 2.44$, p=0.13) or interactions ($F_{1,42} = 0.16$, p=0.69) in the sustained response.

To examine whether the sustained fMRI response was related to autism severity, we correlated ASD individuals' mean sustained response with total scores on the Autism Diagnostic Observation Schedule (ADOS). Sustained response in the auditory ROI during the fixed-interval condition was significantly correlated with ADOS scores ($r = 0.53$; p=0.02; *Figure 3*). Correlations between ADOS scores and the sustained response in the randomized-interval condition ($r = 0.07$; p=0.78) and with both timing conditions in the visual ROI (fixed-interval, $r = 0.28$, p=0.24; randomized-interval, $r = 0.06$, p=0.81) were not statistically significant.

In addition to the sustained response difference between ASD and NT in auditory cortex, we also observed that the fMRI signal 'undershoot' during the post-stimulus period (i.e. during the blank period after the 20 s stimulus block was over) was lower in NT compared to ASD individuals (*Figure 2A and B*). To assess whether these post-stimulus effects played a role in the adaptation results above, we formed matched groups based on the post-stimulus response magnitude in auditory cortex. Specifically, we included an individual subject if their post-stimulus response magnitude (average between 28 and 34 s after block onset) was in a prespecified range. This range was iteratively adapted until there was no group difference in post-stimulus response. Comparing these matched groups, we found that the difference in sustained response in auditory cortex remained in the fixed-interval condition (*Figure 4*; $t_{23} = 1.99$, p=0.058). We also note that the largest post-

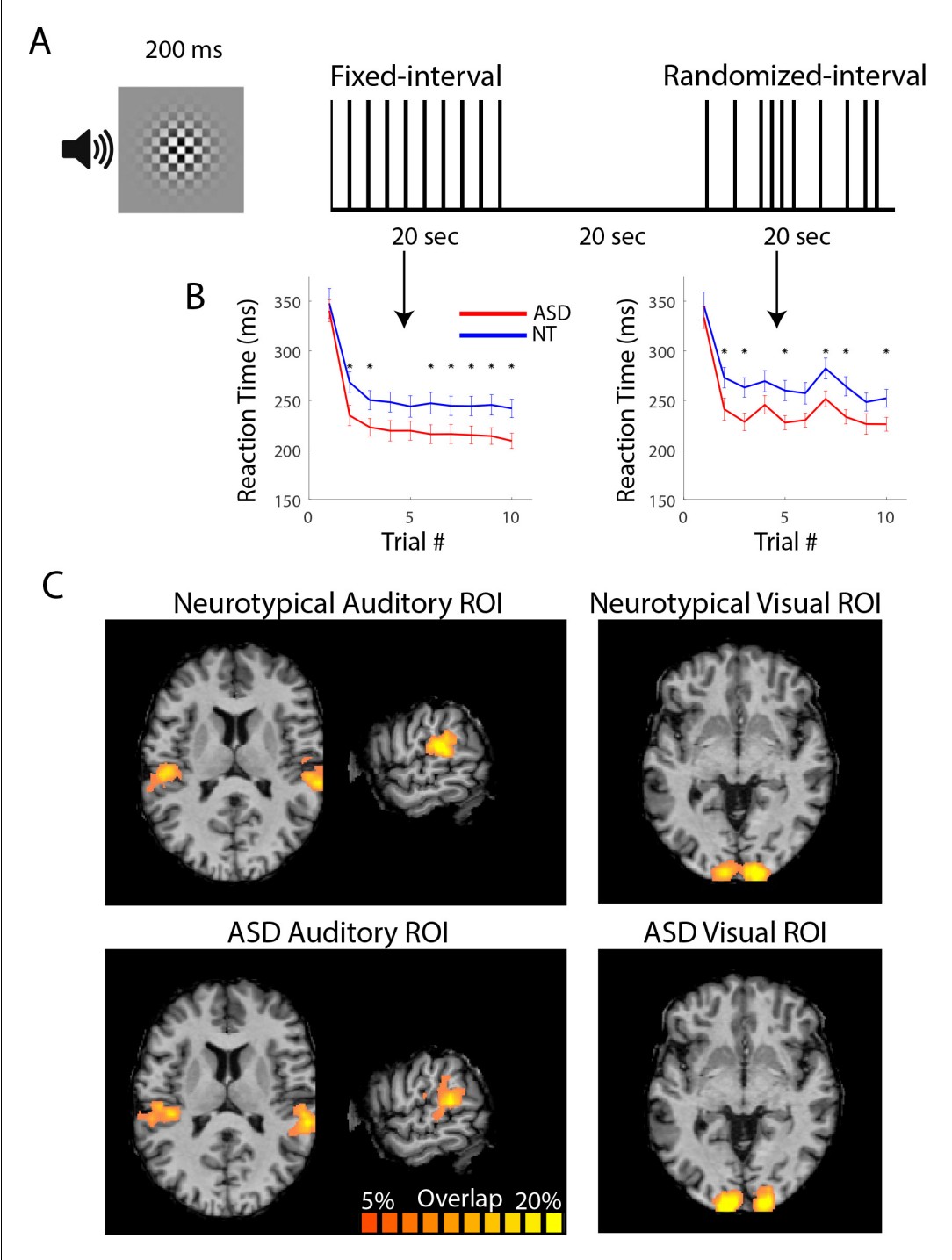

**Figure 1.** Task and ROI selection. (**A**) The stimulus consisted of a checkerboard presented for 200 ms accompanied for the duration by auditory white noise. Subjects were asked to respond with a button press following the stimulus. Stimuli were presented with fixed or randomized inter-stimulus intervals in 20 s blocks. Stimulus blocks alternated with 20 s of fixation. (**B**) Reaction times were shorter for ASD compared to NT participants from the second trial and on. (*)=p < 0.05. (**C**) ROIs in left and right visual and left and right auditory areas were selected based on activation to stimulus versus fixation blocks over the three experimental runs. Probability maps/heat maps showing auditory and visual ROI locations for NT (top) and ASD (bottom) participants. Percent overlap across subjects calculated in Talairach space and displayed on an individual subject's anatomical image.

DOI: https://doi.org/10.7554/eLife.36493.002

The following source data is available for figure 1:

**Source data 1.** Button press reaction times.

DOI: https://doi.org/10.7554/eLife.36493.003

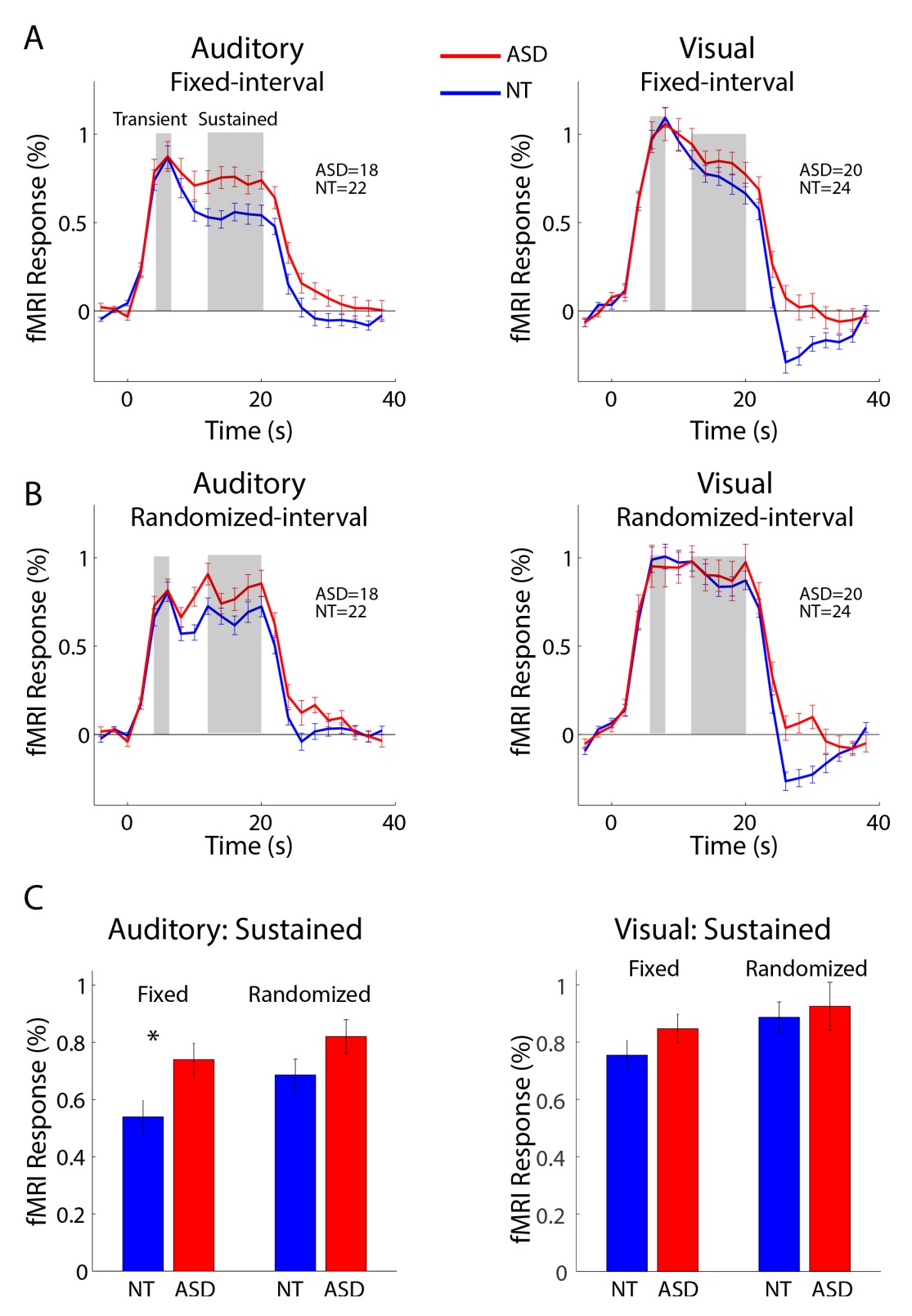

**Figure 2.** (A) Mean fMRI response timecourses in the fixed-interval timing blocks for NT (blue) and ASD (red) groups in the auditory (left, N(NT)=22, N(ASD)=18, and visual (right, N(NT)=24, N(ASD)=20) ROIs. (B) Same as (A), but for the randomized-interval condition. (C) Mean response averaged over the sustained period. Error bars indicate the standard error of the mean.

DOI: https://doi.org/10.7554/eLife.36493.004

*Figure 2 continued*

The following source data is available for figure 2:

**Source data 1.** FMRI timecourses and averaged sustained responses in auditory and visual cortex.
DOI: https://doi.org/10.7554/eLife.36493.005

stimulus difference is in visual cortex and, despite the large difference, the sustained response is equivalent in both groups. Thus, any mechanism that creates a post-stimulus difference, by itself, is not sufficient to also cause a difference in sustained response.

Our fMRI results demonstrate an elevated sustained neural response in auditory cortex in ASD compared to NT that is not present in the visual sustained response. This suggests ASD may be associated with increased sensory behavioral abnormalities compared to NT in the auditory compared to visual domain. To test this prediction, we used the Adult/Adolescent Sensory Profile (AASP (*Brown and Dunn, 2002*)) to assess levels of sensory processing in everyday life in a larger sample that included subjects from the above fMRI experiment, plus an additional 11 adults with ASD and 16 NT (see Materials and methods). The AASP is a 60-item questionnaire based on Dunn's model of sensory processing (*Dunn, 1997*). Relevant for the current study is the model's concept of a neurological threshold: low thresholds reflect tendencies toward sensory avoiding and sensory sensitivity behaviors. Participants answer on a five-point scale (1 = almost never; 5 = almost always). For example, answering with a '5' to items such as 'I stay away from noisy settings' and 'I am distracted if there is a lot of noise around' would be interpreted as reflecting a low neurological threshold in the auditory domain.

Based on our fMRI results, we predicted that because there were elevated auditory neural responses in ASD compared to NT, there would be an increase in everyday behaviors that reflect low auditory sensory thresholds. Consistent with our prediction AASP responses showed a pattern consistent with low auditory threshold for ASD vs. NT individuals (i.e. higher responses on questions associated with sensory sensitivity or sensory avoiding; $t_{78}$ = 3.14, p=0.002, Cohen's d = 0.70). There were no significant group differences in scores associated with a high threshold in the auditory

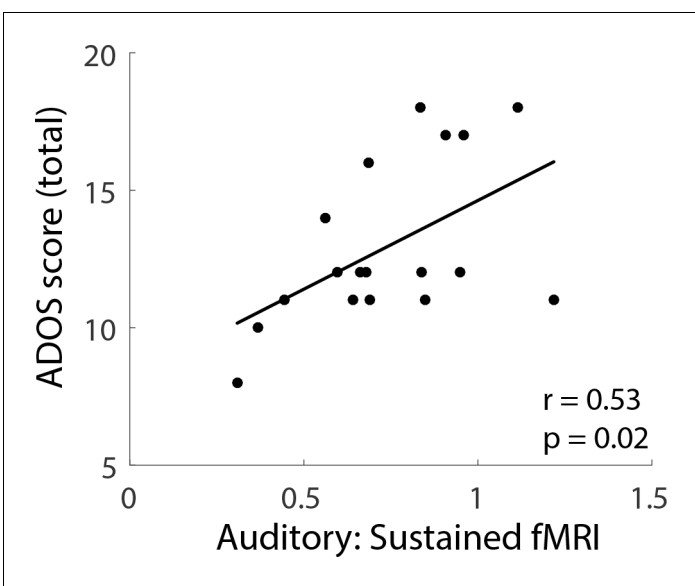

**Figure 3.** Individual differences (ASD participants) in the sustained fMRI response in the auditory cortex fixed-interval condition plotted against total ADOS scores.
DOI: https://doi.org/10.7554/eLife.36493.006

The following source data is available for figure 3:

**Source data 1.** Sustained auditory fMRI response and ADOS scores in the ASD group.
DOI: https://doi.org/10.7554/eLife.36493.007

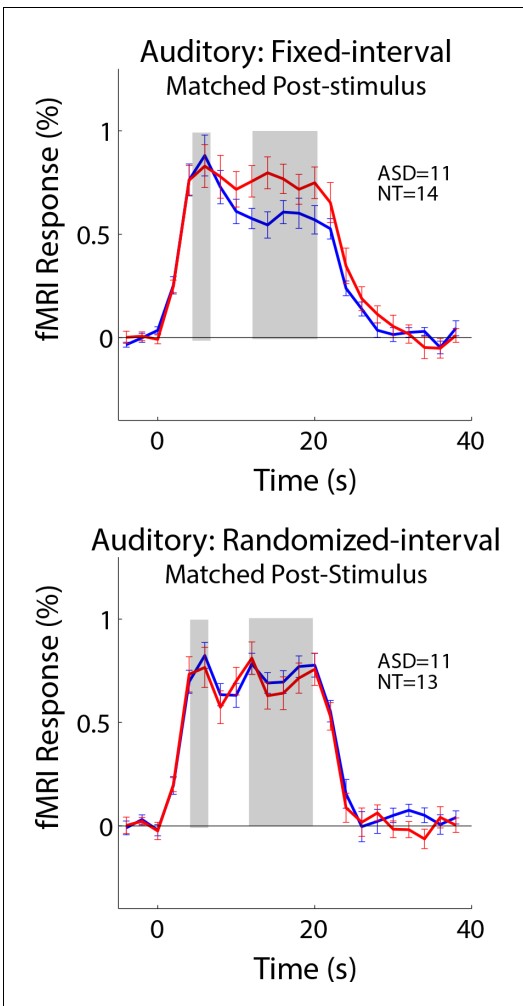

**Figure 4.** Subgroups were formed that were equated for the post-stimulus response. Differences in the sustained response remained in the fixed-interval condition (top).

DOI: https://doi.org/10.7554/eLife.36493.008

The following source data is available for figure 4:

**Source data 1.** FMRI timecourses in auditory cortex after matching for post-stimulus response.

DOI: https://doi.org/10.7554/eLife.36493.009

domain ($t_{78}$ = 0.86, p=0.40, Cohen's d = 0.19) or either a low ($t_{78}$ = 1.56, p=0.12, Cohen's d = 0.34) or high ($t_{78}$ = 1.11, p=0.27, Cohen's d = 0.25) threshold in the visual domain. Although consistent with our prediction, the generalizability of the AASP results should be interpreted with some caution; there were no significant statistical relationships between individual differences in the auditory- and visual-specific sensory processing scores and individual differences in the fMRI sustained response. And when limiting the sample to subjects that participated in the fMRI experiment the group difference in low sensory auditory threshold did not reach statistical significance. However, overall, both the fMRI and behavioral questionnaire findings are consistent with a hypothesis that auditory sensory processing may be more disrupted in individuals with ASD compared to visual sensory processing and reflect elevated neural responses.

## Discussion

Our results demonstrate increased neural responsiveness in ASD that occurs after repeated stimulation, is region specific, and sensitive to the temporal pattern of stimulation. Specifically, we observed a larger fMRI response in auditory cortex in individuals with ASD that begins approximately 12 s after stimulus onset which, after accounting for a hemodynamic delay of ~6 s, corresponds to about 6 s after the beginning of the stimulus block. This means that by the third or fourth stimulus presentation, a difference in response magnitude emerges between ASD and NT participants. Overall, the sustained response in NTs was strongly reduced (67%) compared to the initial transient response. In ASD, however, the response in the sustained period more closely resembled the transient response (88%). The difference between ASD and NT in auditory cortex was statistically significant only when the timing of the stimulus occurred at fixed intervals. In addition, individual differences in sustained response magnitude correlated with ASD severity scores.

The most likely interpretation for these results is that there is reduced auditory adaptation in individuals with ASD compared to NT controls. Adaptation is a neural regulatory process that adjusts neural responses to the current sensory environment and, with very simple stimulus conditions such as those in our experiment, often involves response reductions (*Kohn, 2007*; *Solomon and Kohn, 2014*; *Larsson et al., 2016*). In auditory cortex, a region that is highly sensitive to stimulus temporal structure (*Boemio et al., 2005*), neural adaptation is more likely when the temporal structure of the stimulation occurs at regular intervals. For example, auditory adaptation in response to repeated sounds is larger for expected versus unexpected stimuli (*Todorovic et al., 2011*). We suggest that in NTs, a representation of the regular temporal structure in the fixed-interval timing condition is formed after approximately 3–4 stimulus presentations. This representation leads to a reduction in

responses to subsequent stimulus presentations. This interpretation is most closely associated with fMRI 'repetition suppression' (*Barron et al., 2016*) in that it is sensitive to extracted stimulus features (temporal regularity) and not to other mechanisms such as simple neural fatigue. In ASD, either a representation of the fixed stimulus timing was not as strongly formed, or the representation does not subsequently influence neural responses as strongly. That the neural response in individuals with ASD reflects an inability to extract the temporal regularities of the stimulation sequence is consistent with theoretical accounts that suggest that many of the core symptoms of ASD, including the failure to adapt or habituate, can be framed as an inability to form predictions (*Sinha et al., 2014*) and incorporate prior experiences (*Pellicano and Burr, 2012*). Further supporting this interpretation, in the randomized interval condition where there was no temporal regularity in the stimulation, no significant differences emerged between ASD and NT participants and there was no relationship between the sustained response and ASD symptom severity. We implemented a simple model of adaptation where neural response magnitudes for each stimulus presentation in a fixed-interval block begins to reduce after the second stimulus presentation and asymptotes after the fifth stimulus presentation. ASD and NT model responses only differed in their asymptotic response magnitude (lower for NT than ASD). Convolving these different modeled neural responses with a canonical hemodynamic response function yields a predicted fMRI timecourse that closely resembles our auditory cortex measurements for ASD and NT participants (*Figure 5*), suggesting at least one plausible way in which underlying neural responses might differ between ASD and NT over the course of successive stimulus presentations. Future work that uses methodologies such as ERP that can better isolate individual responses may be useful for testing this adaptation model.

Previous findings that have assessed adaptation (also referred to as 'habituation') in ASD have yielded mixed results. Older findings that relied on physiological measures such as electrodermal (*van Engeland, 1984*) and respiratory/pulse-rate (*James and Barry, 1980*), yielded no definitive conclusion about whether habituation was altered in ASD (*Rogers and Ozonoff, 2005*). However, more recent findings using behavioral measures across a broad range of stimulus types and paradigms (*Tommerdahl et al., 2007*; *Tannan et al., 2008*; *Puts et al., 2014*; *Pellicano et al., 2007*; *Fiorentini et al., 2012*; *Pellicano et al., 2013*; *Turi et al., 2015*; *Noel et al., 2017*; *Mosconi et al., 2013*) have demonstrated more consistent evidence of reduced adaptation in ASD. In addition, recent brain imaging results have shown that subjects with ASD show less reduction in fMRI response than NT subjects to aversive stimulation, measured in successive blocks of presentation (*Green et al., 2015*), however, without a manipulation of stimulus properties it is unclear whether

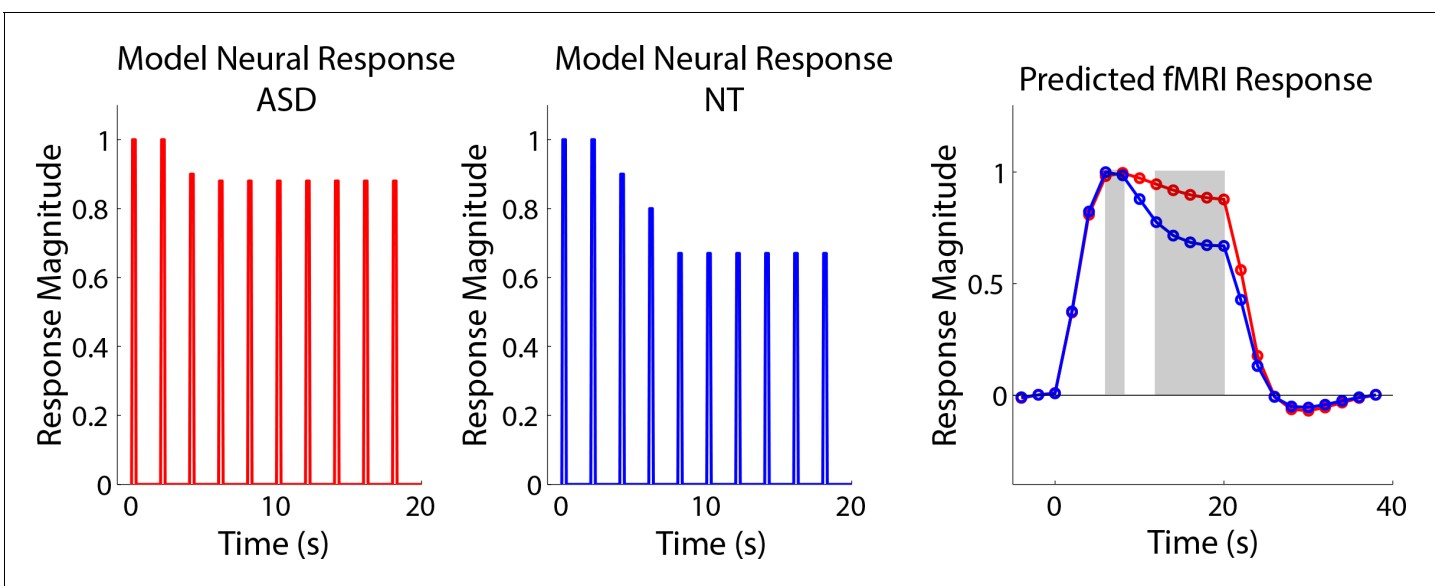

**Figure 5.** Model neural responses that reflected different degrees of adaptation (left = ASD, less adaptation; middle = NT, more adaptation) were convolved with a canonical hrf to produce expected fMRI timecourses for each group (right).
DOI: https://doi.org/10.7554/eLife.36493.010

this effect reflects neural or hemodynamic properties or whether it relies on the same processes as the rapidly appearing differences observed here. In addition, fMRI adaptation paradigms using face stimuli have shown reduced adaptation/habituation in regions such as the amygdala (*Kleinhans et al., 2009*) and fusiform face area (*Ewbank et al., 2017*). In contrast, our results reveal an early sensory disruption of adaptation to non-social stimuli and are consistent with previous ERP findings of reduced habituation to repeated sounds in infants at high risk for autism (*BASIS Team et al., 2011*).

We did not observe a significant difference between ASD and NT participants in the sustained response in visual cortex. This may indicate that disruptions of adaptation in ASD are region-specific. Indeed, a recent investigation revealed equivalent fMRI repetition suppression in the visual system in individuals with ASD (*Utzerath et al., 2018*). However, the contributions of different mechanisms that underlie adaptation (*Lanting et al., 2013*) may vary by cortical region. It is possible that future research that uses timing and/or stimulus parameters that are more specifically tailored to properties of early visual cortical neurons (*Fang et al., 2005*) may reveal differences in early sensory adaptation between ASD and NT individuals. Further, our procedure used a combined auditory-visual stimulus and previous findings have suggested there may be disrupted audio-visual integration in individuals with ASD (*Foss-Feig et al., 2010*; *Kwakye et al., 2011*; *Stevenson et al., 2014*). Thus, future experiments may benefit from separating the stimulus modalities to examine adaptation effects separately in each sensory domain.

To help equate attentional engagement participants were engaged in a simple stimulus-response task that required a button-press to each stimulus presentation. ASD participants responded with shorter RTs and there was no obvious speed-accuracy (i.e. pressing the button more) tradeoff. This result is surprising given recent meta-analyses demonstrating no difference in RT between ASD participants and controls (*Ferraro, 2016*), a conclusion consistent with our observation of no RT differences for first-trials in a block. However, one unique feature of our experiment is its limited cognitive demand and repetitive trial structure which may reveal differences that are specifically relevant for sustained tasks. With respect to our fMRI findings, it is known that motor-related signals can suppress excitatory neurons in auditory cortex (*Schneider et al., 2014*). Thus, an experimental paradigm that removes potential motor influences may be warranted in future experiments.

Sensory symptoms, now included in the diagnostic criteria (*American Psychiatric Association, 2013*), are very common in ASD (*Leekam et al., 2007*; *Brown and Dunn, 2002*; *Rogers and Ozonoff, 2005*; *Ben-Sasson et al., 2009*) and persist across age (*Leekam et al., 2007*), are present in all cognitive abilities (*Leekam et al., 2007*), and have unique features when compared to other neurodevelopmental disorders (*Rogers et al., 2003*). However, sensory symptoms in ASD are also highly heterogeneous (*Crane et al., 2009*) and can include hyperresponsiveness (over-reactivity to sensory stimuli), hyporesponsiveness (under-reactivity to sensory stimuli), and sensation seeking (craving/fascination with certain stimuli) (*Rogers and Ozonoff, 2005*; *Ben-Sasson et al., 2009*). Our assessment of sensory symptoms in everyday life measured with the AASP provides further evidence that sensory abnormalities are strongly associated with ASD. Overall, we found that the sensory abnormalities are larger in the auditory than visual domain and specifically reflect low sensory threshold behaviors. Overall, our findings reveal a marked elevation in responsiveness in auditory cortex in ASD that may contribute to the complex sensory symptomatology that is associated with the disorder.

## Materials and methods

### Subjects

Twenty-four subjects with autism spectrum disorder (ASD; 16 males; 22 right-handed) and 29 neurotypical (NT) subjects (14 males; 29 right-handed) participated in the experiment. There were multiple exclusionary criteria that were applied to each subject's fMRI data to determine the final total number of subjects and ROIs. First, some subjects were excluded outright before analysis due to observed data quality in the scanner (for example, falling asleep, not performing the task, or multiple head movements > 2 mm). This eliminated four subjects (2 ASD, 2 NT), leaving 22 ASD and 27 NT subjects. Second, we applied a data cleaning procedure (explained below) based on head motion and/or poor behavioral performance during a scan. This removed one additional ASD subject,

leaving 21 ASD and 27 NT subjects. Third, two ASD subjects did not have definable auditory ROIs (identification procedures described below). Fourth, we excluded subjects based on excessively noisy timecourses as assessed with a power spectrum test (explained below) on a per ROI basis. This removed one additional ASD and five NT subjects in the auditory ROI; it removed one additional ASD and 3 NT subjects in the visual ROI. Overall, data were retained for 18 ASD and 22 NT subjects in the auditory ROI and 24 and 20 in the visual ROI (see *Table 1*). This sample size is similar to those reported in relevant, methodologically similar recent studies (*Dinstein et al., 2012*; *Brieber et al., 2010*).

All subjects had normal IQ (WASI-II Full Scale IQ of at least 80), and normal or corrected-to-normal vision. Groups were of equal ages and IQ (mean IQ of subjects with autism: 114; NT subjects: 112; $t_{45}$ = 0.39, p=0.70; mean age of subjects with autism: 23 years; NT subjects: 24 years; $t_{45}$ = −0.96, p=0.34). In addition to the subjects reported above who completed the fMRI experiments, an additional 11 ASD subjects (i.e. a total of 35; 24 males) and an additional 16 NT (i.e. total of 45; 28 males) completed the Adult/Adolescent Sensory Profile as part of a separate, ongoing experiment. We report results for all subjects who completed the questionnaire. This larger sample was also of equal IQ and age (mean IQ of subjects with autism: 107; NT subjects: 112; $t_{75}$ = 1.32, p=0.19; mean age of subjects with autism: 23 years; NT subjects: 23 years; $t_{75}$ = −0.10, p=0.91). All subjects provided written informed consent to participate. The Institutional Review Board of the University of Washington (UW) approved the research protocol. Subjects with ASD met diagnostic criteria for ASD on the Autism Diagnostic Interview-Revised (ADI-R (*Rutter et al., 2003*)), the Autism Diagnostic Observation Schedule—2nd Edition (ADOS-2 *Lord et al., 2012*) and according to expert clinical judgment using DSM-5 *American Psychiatric Association, 2013* criteria.

## MRI acquisition

Scans were acquired with a Philips Achieva 3T MRI system. A T1-weighted structural scan was acquired at the beginning of the scan session, followed by three functional gradient-echo EPI scans with axial orientation (30 slices with 3 mm inplane resolution and 0.5 mm gap, 2 s TR, 25 ms TE, 79° flip angle, A-P phase-encode direction). A single TR EPI scan with opposite phase-encoding direction (P-A), but otherwise identical to those above, was acquired for use in correcting geometric distortions. Each subject underwent a single scanning session, lasting approximately one hour (the scan session also included acquisition of spectroscopy data for a separate experiment). When possible, subjects' eyes were tracked during scanning using an Eyelink 1000 Plus eyetracker, sampling at 1000 Hz. Due to the challenges of eyetracking in the scanner, we were able to collect data for 55% of subjects (N(NT)=17, N(ASD)=9). We found no group differences in eye movement behavior in the subjects whose eyes were successfully tracked. We found no difference in the proportion of time spent fixating ($t_{24}$ = −0.62, p=0.54), or the mean ($t_{24}$ = −1.18, p=0.25) or standard deviation ($t_{24}$ = −1.20, p=0.24) of the distance of eye position from the fixation mark.

## Stimuli

Stimuli were presented using Presentation software running on a Windows XP computer. Images were projected onto a screen behind the subject's head via either an Epson Powerlite 7250 or an

**Table 1.** Subject enrollment numbers and exlucsions.

|  | ASD |  | NT |  |
| --- | --- | --- | --- | --- |
| Total enrolled | 24 |  | 29 |  |
| Pre-analysis exclusions (sleep, large head motions, etc.) | 2 |  | 2 |  |
| Data cleaning exclusion | 1 |  | 0 |  |
|  | Auditory | Visual | Auditory | Visual |
| ROI definition exclusion | 2 | 0 | 0 | 0 |
| Power spectrum exclusion | 1 | 1 | 5 | 3 |
| Total Usable Data | 18 | 20 | 22 | 24 |

DOI: https://doi.org/10.7554/eLife.36493.011

Eiki LCXL100A projector (following a hardware failure), both operating at 60 Hz, and with linearized luminance profiles. Subjects viewed the projected images using a mirror positioned above their eyes, for an effective viewing distance of 66 cm. Sound was delivered at 44.1 kHz using MRI compatible earbuds (S14, Sensimetrics). Subjects wore protective ear muffs over the earbuds to attenuate acoustic noise from the scanner. Before scanning, subjects verified that the auditory stimulus was presented at an audible and comfortable volume.

The stimulus consisted of a Gaussian-windowed (with FWHM of 2.75 deg. visual angle; approximate visible size of 5.2 deg), full-contrast checkerboard (check size of 0.4 deg) image presented on a uniform background accompanied by audio white noise. Visual and auditory stimuli were presented simultaneously for 200 ms in a blocked design, with the stimulus presented 10 times within each stimulus block. Blocks were of two types: fixed-interval and randomized-interval. In fixed-interval blocks, the stimuli were separated by 1800 ms of rest, resulting in a stimulus presentation every 2 s. In randomized-interval blocks, the inter-stimulus interval was a random value drawn from a uniform distribution bounded by 800 ms and 2800 ms. Stimulus blocks alternated with rest blocks. A fixation cross appeared at the center of the display whenever the stimulus was off. Each subject completed three runs of eight stimulus blocks and nine rest blocks each. The protocols (stimulus timing) for these runs were identical across all subjects, but the variable blocks differed across the three fMRI scans. Subjects were instructed to use the index finger of their dominant hand to press a button as quickly as possible following the appearance of each stimulus. Subjects completed one practice run during a mock scan session prior to scanning.

## MRI data preprocessing

Data was preprocessed using BrainVoyager (Brain Innovation, Maastricht, The Netherlands) software. EPI data was motion-corrected, corrected for distortion due to magnetic field inhomogeneities, high-pass filtered (cutoff = 2 cycles/scan), and coregistered to the AC-PC-aligned T1 structural scan. To identify regions of interest corresponding to early visual and auditory cortical areas, statistical activation maps were determined from z-transformed data from all three functional runs. Fixed- and randomized-interval conditions were treated as a single condition (versus rest) to generate a boxcar predictor, which was then convolved with a double gamma HRF. This predictor plus the three translation and three rotation parameters obtained during motion correction were used as the design matrix in a general linear model that was fit to the timecourse of each voxel. The resulting activation maps from the t-statistic for the model fit were initially thresholded at $p<0.05$ (Bonferroni corrected). ROIs were selected manually from the most significant areas of activation near visual (left and right) and auditory (left and right) cortices, yielding 4 ROIs for each subject. If multiple activation clusters were present, ROIs were selected according to the following criteria. For the early visual ROI, the cluster nearest the occipital pole, in line with the calcarine sulcus, was selected. The anatomical location of auditory activation was variable across subjects; the most significantly activated cluster on the superior temporal lobe was selected. The 20 most significantly activated voxels in the cluster defined the final ROI in each region. For some ROIs in some subjects, fewer than 20 voxels met the threshold criteria; in these cases, the threshold was relaxed until 20 voxels could be selected. If no obvious cluster of voxels was present after the threshold was lowered, the ROI was excluded. This resulted in two ASD subjects without an identifiable auditory ROI. All subjects had an identifiable visual ROI.

We examined the robustness of the fMRI response used to define the ROIs for the two groups. As explained above, each subject had one activation map that was used to determine ROI positions. For each subject, we took the average F-value (an index of 'response robustness') of the activation map in each ROI location. Then, we performed a 2-way group x ROI ANOVA. There was no main effect of group ($F_{1,43}$= 0.016, p=0.90), a main effect of ROI ($F_{1,43}$ = 32.43, p<0.0001), and no interaction ($F_{1,43}$ = 3.027, p=0.09). The main effect of ROI was in a direction that matched our subjective impression – responses were stronger in visual than auditory cortex. Repeating this analysis using the maximum F-value in each ROI as the index of response robustness (instead of the mean) yielded the same statistical outcome. Overall, the signals used to define the ROIs were equally robust in the two groups.

## fMRI data analysis

Average timecourses across the 20 voxels in each ROI were determined for each run. Percent-transformed timecourses were then calculated for each block condition. First, for each stimulus block, we extracted 22 timepoints corresponding to −4 s before stimulus onset to 38 s after stimulus onset (TR, the sampling rate, was 2 s). Then we converted the values to percent signal change relative to the mean value of timepoints −4,−2, and 0 (reflecting the best-estimate of 'baseline' before stimulus onset). Specifically, each timecourse was normalized by subtracting and dividing by the mean of the pre-stimulus TRs and multiplying by 100. The resulting block timecourses were then averaged over all blocks of the same condition, yielding one such timecourse for randomized-interval timing blocks and another for fixed-interval blocks. Blocks that did not meet the criteria detailed below for head motion and task performance were excluded prior to averaging. Data for a given block was excluded due to head motion if the subject's head moved more than 0.9 mm between two successive TRs (frame-wise displacement >0.9 mm [*Siegel et al., 2014*]) up to and including 8 TRs before or 1 TR after the stimulus block. A block was excluded on the basis of task performance if it contained any misses (failure to press the button after a stimulus appearance) or more than one false alarm (more than one button press after appearance of a stimulus). If more than half the blocks of either condition were excluded, or a subject had more than seven behavioral errors in a run, the entire run was excluded. If two out of three runs were excluded, all data for the subject were excluded. Seven out of 147 total runs (1 NT and 6 ASD) and one complete subject (ASD) were excluded using this procedure. Noisy data were further removed by excluding the resulting timecourses for which less than 50% of the power (as per discrete Fourier analysis of the ROI timecourse) was at one cycle per duration of the extracted event-related timecourse (44 s). This resulted in the exclusion of 1 additional ASD subject, and 14 of the remaining subjects retaining data for only one or two of the two ROIs. With cleaning, data was retained for 22 NT and 18 ASD subjects in the auditory ROI and 24 and 20 in the visual ROI.

## Adult/adolescent sensory processing (AASP)

The Adult/Adolescent Sensory Profile (*Brown and Dunn, 2002*) is a 60-item questionnaire probing sensory behaviors in everyday life. Each item asks how often the respondent performs a particular behavior with answers 'Almost Never', 'Seldom', 'Occasionally', 'Frequently' and 'Almost Always' scored 1, 2, 3, 4 and 5 points, respectively. Based on our fMRI findings, we *a priori* selected items on the questionnaire limited to audition and vision. This included 10 vision-related questions and 11 auditory-related questions. In addition to sensory modality, each question is identified as belonging to one of four hypothesized sensory dimensions: (1) low registration, (2) sensation seeking, (3) sensory sensitivity, and (4) sensory avoiding. Dimensions 1 and 2 relate to a high sensory threshold and dimensions 3 and 4 relate to a low sensory threshold. Based on the fMRI findings, we *a priori* expected a change in behaviors related to sensory thresholds. We, therefore, combined scores on dimensions 1 and 2 and dimensions 3 and 4.

## Statistics

All statistical analyses were performed in Matlab 2013b or SPSS 19. Data were visually inspected to ensure they were normally distributed and we used standard parametric tests to determine statistical significance. Two-tailed t-tests were used to test for differences between ASD and NT groups, and correlation was quantified with Pearson's r.

## Additional information

### Funding

| Funder | Grant reference number | Author |
| --- | --- | --- |
| National Eye Institute | F32 EY025121 | Michael-Paul Schallmo Scott Murray |
| National Institute of Mental Health | R01 MH106520 | Raphael A Bernier Scott Murray |

| National Eye Institute | T32 EY007031 | Rachel Millin
Michael-Paul Schallmo |

The funders had no role in study design, data collection and interpretation, or the decision to submit the work for publication.

## Author contributions

Rachel Millin, Conceptualization, Software, Formal analysis, Investigation, Visualization, Methodology, Writing—original draft, Writing—review and editing; Tamar Kolodny, Conceptualization, Software, Formal analysis, Investigation, Methodology, Writing—review and editing; Anastasia V Flevaris, Michael-Paul Schallmo, Conceptualization, Software, Funding acquisition, Investigation, Methodology, Writing—review and editing; Alexander M Kale, Software, Investigation, Methodology, Writing—review and editing; Jennifer Gerdts, Resources, Data curation, Investigation, Writing—review and editing; Raphael A Bernier, Conceptualization, Resources, Supervision, Funding acquisition, Writing—review and editing; Scott Murray, Conceptualization, Resources, Software, Formal analysis, Supervision, Funding acquisition, Investigation, Methodology, Writing—original draft, Project administration, Writing—review and editing

## Author ORCIDs

Rachel Millin http://orcid.org/0000-0001-8555-3496
Tamar Kolodny http://orcid.org/0000-0002-0163-1766
Alexander M Kale http://orcid.org/0000-0001-7668-2800
Michael-Paul Schallmo http://orcid.org/0000-0001-8252-8607
Scott Murray http://orcid.org/0000-0003-3241-3646

## Ethics

Human subjects: All subjects provided written, informed consent prior to participating. Research protocols were approved by the Institutional Review Board of the University of Washington (protocol # 556), and conformed to the ethical principles of the Declaration of Helsinki for research involving human subjects.

## Decision letter and Author response

Decision letter https://doi.org/10.7554/eLife.36493.014
Author response https://doi.org/10.7554/eLife.36493.015

# Additional files

## Supplementary files

• Transparent reporting form
DOI: https://doi.org/10.7554/eLife.36493.012
Source data files have been provided for Figures 1, 2, 3, and 4.

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
