## [Decision Letter]

Thank you for submitting your article "Disrupted neural adaptation in autism spectrum disorder" for consideration by *eLife*. Your article has been reviewed a Senior Editor, Richard Ivry, a Reviewing Editor, Barbara Shinn-Cunningham, and three reviewers. The following individuals involved in review of your submission have agreed to reveal their identity: Sarah Haigh (Reviewer #2); Mark Wallace (Reviewer #3).

The reviewers have discussed the reviews with one another and the Reviewing Editor has drafted this decision to help you prepare a revised submission.

Summary:

This manuscript describes an fMRI experiment to answer the question whether neural adaptation is reduced in people with autism spectrum disorder (ASD) compared to neurotypical individuals. Participants were presented with an audio-visual stimulus (checkerboard + white noise) either isochronously every 2 seconds (regular) or with a variable onset-to-onset interval. Participants were asked to press a button after each audio-visual stimulus. The authors find that fMRI BOLD signal is less reduced with sustained stimulation in people with ASD in motor, auditory, and visual regions.

All of the reviewers agree that this study would be of interest to *eLife* readers. The papers findings fit with a growing view that reduced adaptation may lead to impaired processing (e.g., in people with hearing loss or people with schizophrenia). However, the three reviewers agree that the conclusions regarding altered "excitability" in ASD should be toned down and alternatives considered, as elaborated below. In addition, the manuscript treats the somato-motor, auditory, and visual results as parallel, when they are not. The Introduction, Results, and Discussion all need to be altered to give a more balanced view. All of the reviewers' points have been compiled into the list, below.

Essential revisions:

None of the major issues undercut the basic results; however, each needs to be acknowledged and incorporated into how the manuscript presents and discusses the findings.

1) To address neural excitability directly, a different recording technique such as EEG or MEG would have been a better choice than fMRI, which tracks metabolic activity rather than neural spiking. While the results are consistent with changes in neural excitability in ASD compared to neurotypical controls, they do not prove that this is the reason for the differences in BOLD activation.

2) Related to this, the authors argue that neural adaptation is a domain general trait, based on the correlation of adaptation between cortical areas. However, changes in neural excitability are local phenomena and the time over with which neurons recover from adaptation likely is modality specific (i.e., time constants of neurons differ). The fact that adaptation correlates across regions may reflect a more cognitive correlate related to repetition suppression (e.g., differences in prediction, or differences in vigilance, with ASD participants treating most stimuli as novel), or some other domain-general mechanistic process. A fairer discussion is needed, which discusses repetition suppression from an fMRI perspective rather than attributing differences to "neural excitability."

3) Adaptation is commonly investigated using sensory stimulation. Here, each stimulus presentation was accompanied by a button press and evaluated motor region responses. It is difficult to understand the logic of relating adaptation in motor regions driven by a motor task to adaptation of sensory responses in auditory and visual areas. Why didn't you evaluate responses to tactile stimulation, or even better, sensory stimulation presented independently for each modality? Given the methodology, it is not surprising that reaction times and motor region responses are correlated strongly. With a button press after each stimulus, participants likely engage rhythmic expectations, motor anticipation, attention, and other cognitive processes. It is then hard to know the cause underlying of the observed effects. These concerns limit what conclusions can be drawn.

4) The data show that fMRI BOLD signal is less reduced with sustained stimulation in people with ASD in motor and auditory regions. However, the effect is not strong in visual cortex; moreover, the temporal dynamics of the adaptation effects differ in visual cortex (i.e., it seems to be occurring later). From the animal literature, we know that movements lead to a suppression in auditory regions and an increase in visual regions (e.g., Schneider et al., 2014, Nature). Relatedly, the visual ROIs also do not significantly correlate with RT (supplementary materials). While the performed ANOVA does not show a significant interaction between group and ROI, it does not address whether there a significant three-way interaction. Including effect sizes of the group difference in adaptation will allow the reader to get a better sense of how domain general the results truly are.

5) The adaptation metric (Figure 1) seems reduced across the board in ASD (although the effects are only significant for the regular, not the variable, condition). The introduction of the integration metric is questionable. First, what is it intended to represent? Second, by the computations, the IM is directly related to the AM. Specifically, AM = (Initial – Sustained) / Initial, while the IM is simply a weighted version of (Initial + Sustained) / Initial. AM and IM are not independent. Mathematically, one expects them to be negatively correlated, as they are. This analysis should be removed.

6) The finding of speeded RTs in response to the combined AV stimulus in the ASD group is surprising given that others have failed to see multisensory-mediated improvements on similar tasks in ASD. How do the authors reconcile these results with the broader literature?

7) The data cannot be used to decide whether the effects are sensory-mediated or sensorimotor. A control case of a purely passive paradigm (i.e., just presentation of the audiovisual stimulus) without the motor response would have helped isolate the effects.

8) It is difficult to reconcile the mean AM values of individual subjects shown in Figure 3B, which seem to cluster around 0, with the data in Figure 1, where the mean AM value in somato-motor cortex for the regular stimuli is around 0.2. Is there a mathematical error somewhere?

9) Given the nature of the task (a speeded response to a multisensory [audiovisual] target), the observed effects could be due to differences in how the visual and auditory signals are integrated. There is a growing literature illustrating changes in multisensory function in those with autism (see Stevenson et al., 2014).

10) It would be good to point out that future work could focus on the time course of the adaptation to improve our understanding of what underlying mechanistic processes lead to the observed effects. Of course, as an fMRI study, the current work is limited in terms of what it can discern in the temporal domain.

11) Was any effort made to relate individual differences in adaptation to individual differences in clinical features? It would be of great interest to know whether the measured signal adaptation related to clinical metrics such as restricted interests and repetitive behaviors, which would ground the results in real-world characteristics of ASD.

---

## [Author Response]

Summary:All of the reviewers agree that this study would be of interest to eLife readers. The papers findings fit with a growing view that reduced adaptation may lead to impaired processing (e.g., in people with hearing loss or people with schizophrenia). However, the three reviewers agree that the conclusions regarding altered "excitability" in ASD should be toned down and alternatives considered, as elaborated below. In addition, the manuscript treats the somato-motor, auditory, and visual results as parallel, when they are not. The Introduction, Results, and Discussion all need to be altered to give a more balanced view. All of the reviewers' points have been compiled into the list, below.

We are pleased that the reviewers feel our findings are appropriate for *eLife*. Given the overall positive reviews, it may come as a surprise that we have substantially revised the manuscript beyond what was requested in the reviews. This substantial revision process began with an exploration of one of the issues about the late negative response in the visual ROI. On our end, that initiated multiple modeling exercises, an exploration of BOLD ‘undershoot’ effects, a careful consideration of using “metrics” (e.g., relative fMRI values) vs. raw values, and whether it was appropriate to compare ASD and NT fMRI adaptation effects if there were overall magnitude differences in the initial transient response. In the end, and after considerable revision, we feel that the manuscript is substantially improved. Below, we summarize the main differences of this revised manuscript. Then, we respond to each point raised in the initial review. Some of these points are no longer relevant in the revised manuscript.

Summary of changes:

1) New title reflects that the main effect we observed was in auditory cortex.

2) We removed the analysis of the somato-motor ROI. Part of the justification of its removal is theoretical (see reviewer point 3: we agree that comparing motor effects with sensory effects is not well motivated). The other part of the justification was that there was a large difference in the transient response in the somato-motor ROI between NT and ASD. Given this initial transient response difference, it is more difficult to interpret post-transient (adaptation) effects. For example, it is not clear the extent to which a difference in sustained response is ‘inherited’ from a difference in transient response.

3) We removed all analyses involving “metrics” (e.g., adaptation metric). Instead, we present only raw fMRI signal magnitudes in a temporal window where we expect to observe adaptation (“sustained response”, in the new version). This analysis is more straightforward and is especially easy to interpret because there are no differences in any conditions or ROIs in the transient response.

4) We changed the terminology of the timing conditions from “regular” vs. “variable” to “fixed-interval” vs. “randomized-interval”. This more accurately describes how the two conditions differ (i.e., in their inter-stimulus timing structure).

5) We do not directly compare fMRI response magnitudes or response characteristics between the fixed-interval and randomized-interval conditions. Our modeling demonstrated that this comparison is not appropriate. Instead, we present a comparison between ASD vs. NT in each condition separately. We do not feel that this limits any of the conceptual conclusions of the paper.

6) We added an analysis that considers post-stimulus “undershoot” effects. This goes back to the original point about the negative response in the visual ROI.

7) We added an analysis that correlates the fMRI results with ADOS scores (relevant to a specific reviewer point).

8) We added an analysis of sensory symptoms using the Sensory Profile questionnaire.

9) Specifically, with regard to the above reviewer summary, we have replaced the term “excitability” with “responsiveness” as a more general term to refer to the overall amount of neural activity (addressed in reviewer point 1, below).

Essential revisions:None of the major issues undercut the basic results; however, each needs to be acknowledged and incorporated into how the manuscript presents and discusses the findings.1) To address neural excitability directly, a different recording technique such as EEG or MEG would have been a better choice than fMRI, which tracks metabolic activity rather than neural spiking. While the results are consistent with changes in neural excitability in ASD compared to neurotypical controls, they do not prove that this is the reason for the differences in BOLD activation.

This is a good point. Upon consideration, we believe there are multiple reasons to use a term other than “excitability” when referring to our experiments and have replaced it with the more general term “responsiveness”. There are at least two issues with the term “excitability”. First, as the reviewers note, we used fMRI – a metabolic measure – and excitability is a measure related to neural spiking. While there are multiple findings that appear to show a strong link between fMRI response magnitude and spike rate, there are clearly non-spiking processes that affect the fMRI signal. Second, any population-based measure (fMRI, EEG, etc.) cannot specifically examine excitability, per se. Larger amplitude responses in a population-based measure could result from a variety of factors such as more neural excitability, more neurons responding in the population, longer duration responses (fMRI), more synchrony (EEG), shorter refractory times, etc. We feel that “responsiveness” captures the notion of overall more activity without additional speculation about the underlying single-unit mechanisms.

2) Related to this, the authors argue that neural adaptation is a domain general trait, based on the correlation of adaptation between cortical areas. However, changes in neural excitability are local phenomena and the time over with which neurons recover from adaptation likely is modality specific (i.e., time constants of neurons differ). The fact that adaptation correlates across regions may reflect a more cognitive correlate related to repetition suppression (e.g., differences in prediction, or differences in vigilance, with ASD participants treating most stimuli as novel), or some other domain-general mechanistic process. A fairer discussion is needed, which discusses repetition suppression from an fMRI perspective rather than attributing differences to "neural excitability."

We agree that it is unclear to what degree neural changes in ASD are pervasive and domain-general or are confined to specific regions. Moreover, in our revised analyses the change in responsiveness is confined to auditory cortex, and this is now emphasized in the manuscript. We also now refer in the discussion to the possibility that neural mechanisms underlying adaptation effects may differ across cortical regions. Further, we agree with the suggestion that our findings are closely related to “repetition suppression”, and this issue is now discussed in detail in the Discussion section.

3) Adaptation is commonly investigated using sensory stimulation. Here, each stimulus presentation was accompanied by a button press and evaluated motor region responses. It is difficult to understand the logic of relating adaptation in motor regions driven by a motor task to adaptation of sensory responses in auditory and visual areas. Why didn't you evaluate responses to tactile stimulation, or even better, sensory stimulation presented independently for each modality? Given the methodology, it is not surprising that reaction times and motor region responses are correlated strongly. With a button press after each stimulus, participants likely engage rhythmic expectations, motor anticipation, attention, and other cognitive processes. It is then hard to know the cause underlying of the observed effects. These concerns limit what conclusions can be drawn.

This is a fair point. We agree that there are aspects to the design that could be improved to specifically assess sensory adaptation – removing the motor component and restricting it to tactile stimulation, independently stimulating each modality, etc. As discussed in our reviewer response introduction, we agree with the reviewers that the analysis of the somato-motor ROI and its comparison to the sensory areas (auditory and visual) is unwarranted and has been removed in the revision.

4) The data show that fMRI BOLD signal is less reduced with sustained stimulation in people with ASD in motor and auditory regions. However, the effect is not strong in visual cortex; moreover, the temporal dynamics of the adaptation effects differ in visual cortex (i.e., it seems to be occurring later). From the animal literature, we know that movements lead to a suppression in auditory regions and an increase in visual regions (e.g., Schneider et al., 2014, Nature). Relatedly, the visual ROIs also do not significantly correlate with RT (supplementary materials). While the performed ANOVA does not show a significant interaction between group and ROI, it does not address whether there a significant three-way interaction. Including effect sizes of the group difference in adaptation will allow the reader to get a better sense of how domain general the results truly are.

This is one area where the revised version is substantially different. We now discuss the results of the visual and auditory ROIs separately. And, it is very clear that there are large adaptation effects in the auditory ROI but not in the visual ROI. This is evident in the basic fMRI findings but also in the correlation between fMRI and ADOS scores (significant for auditory but not visual ROIs). Also, we now briefly discuss the Schneider finding in relation to future experiments (in the Discussion section). Also, we have added effect sizes to our results.

5) The adaptation metric (Figure 1) seems reduced across the board in ASD (although the effects are only significant for the regular, not the variable, condition). The introduction of the integration metric is questionable. First, what is it intended to represent? Second, by the computations, the IM is directly related to the AM. Specifically, AM = (Initial – Sustained) / Initial, while the IM is simply a weighted version of (Initial + Sustained) / Initial. AM and IM are not independent. Mathematically, one expects them to be negatively correlated, as they are. This analysis should be removed.

All use of metrics have been removed. All data are analyzed as fMRI response magnitude in different temporal windows (transient and sustained).

6) The finding of speeded RTs in response to the combined AV stimulus in the ASD group is surprising given that others have failed to see multisensory-mediated improvements on similar tasks in ASD. How do the authors reconcile these results with the broader literature?

We were surprised to see this as well and do not have a good explanation for the difference. In the revised version we now include a paragraph in the Discussion section that discusses RT differences. We can only speculate that the difference in RTs is related to the simplicity of the task and its sustained nature. But, mechanistically, we do not have a good explanation.

7) The data cannot be used to decide whether the effects are sensory-mediated or sensorimotor. A control case of a purely passive paradigm (i.e., just presentation of the audiovisual stimulus) without the motor response would have helped isolate the effects.

This is true. The potential interaction between motor areas and sensory areas cannot be ruled out as a potential mechanism. However, we do not feel that a passive paradigm is the appropriate solution as this would not be able to rule out the contribution of differential attentional engagement. A possible hybrid design – for example, sensory stimulation where a very occasional target stimulus is presented – might be the most appropriate. We have now included discussion of this issue, suggesting that future experiments should control for potential motor-related contributions.

8) It is difficult to reconcile the mean AM values of individual subjects shown in Figure 3B, which seem to cluster around 0, with the data in Figure 1, where the mean AM value in somato-motor cortex for the regular stimuli is around 0.2. Is there a mathematical error somewhere?

This issue is no longer relevant in the revised manuscript.

9) Given the nature of the task (a speeded response to a multisensory [audiovisual] target), the observed effects could be due to differences in how the visual and auditory signals are integrated. There is a growing literature illustrating changes in multisensory function in those with autism (see Stevenson et al., 2014).

This point is now discussed in the Discussion section.

10) It would be good to point out that future work could focus on the time course of the adaptation to improve our understanding of what underlying mechanistic processes lead to the observed effects. Of course, as an fMRI study, the current work is limited in terms of what it can discern in the temporal domain.

We strongly agree. In our revision, we speculate on the time course of adaptation, to the extent possible with fMRI (Discussion section; Figure 5). We also suggest that ERP would be appropriate for future testing of specific adaptation models.

11) Was any effort made to relate individual differences in adaptation to individual differences in clinical features? It would be of great interest to know whether the measured signal adaptation related to clinical metrics such as restricted interests and repetitive behaviors, which would ground the results in real-world characteristics of ASD.

This is a great suggestion. We added two analyses that relate to clinical/behavioral features. First, we analyzed the fMRI effects and their relationship to ADOS scores. A significant correlation was found between ADOS scores and the fMRI sustained response in auditory cortex in the fixed-interval condition (the only fMRI condition that differed between NT and ASD). Second, we’ve added an analysis of Sensory Profile scores. This analysis showed a difference between ASD and NT in everyday sensory behaviors that reflect a low sensory threshold in the auditory domain.